# A Quality-based Syntactic Template Retriever
# for Syntactically-controlled Paraphrase Generation

**Xue Zhang, Songming Zhang, Yunlong Liang, Yufeng Chen,**[*]
**Jian Liu, Wenjuan Han, Jinan Xu**
Beijing Key Lab of Traffic Data Analysis and Mining,
Beijing Jiaotong University, Beijing, China
{xue_zhang,smzhang22,yunlongliang,chenyf,jianliu,wjhan,jaxu}@bjtu.edu.cn

## Abstract

Existing syntactically-controlled paraphrase generation (SPG) models perform promisingly with human-annotated or well-chosen syntactic templates. However, the difficulty of obtaining such templates actually hinders the practical application of SPG models. For one thing, the prohibitive cost makes it unfeasible to manually design decent templates for every source sentence. For another, the templates automatically retrieved by current heuristic methods are usually unreliable for SPG models to generate qualified paraphrases. To escape this dilemma, we propose a novel Quality-based Syntactic Template Retriever (QSTR) to retrieve templates based on the quality of the to-be-generated paraphrases. Furthermore, for situations requiring multiple paraphrases for each source sentence, we design a Diverse Templates Search (DTS) algorithm, which can enhance the diversity between paraphrases without sacrificing quality. Experiments demonstrate that QSTR can significantly surpass existing retrieval methods in generating high-quality paraphrases and even perform comparably with human-annotated templates in terms of reference-free metrics. Additionally, human evaluation and the performance on downstream tasks using our generated paraphrases for data augmentation showcase the potential of our QSTR and DTS algorithm in practical scenarios.

## 1 Introduction

Paraphrase generation (PG) (Madnani and Dorr, 2010) is to rephrase a sentence into an alternative expression with the same semantics, which has been applied to many downstream tasks, such as question answering (Gan and Ng, 2019) and dialogue systems (Jolly et al., 2020; Gao et al., 2020; Panda et al., 2021; Liang et al., 2019, 2021, 2022). On this basis, to improve the syntactic diversity of paraphrases, syntactically-controlled paraphrase

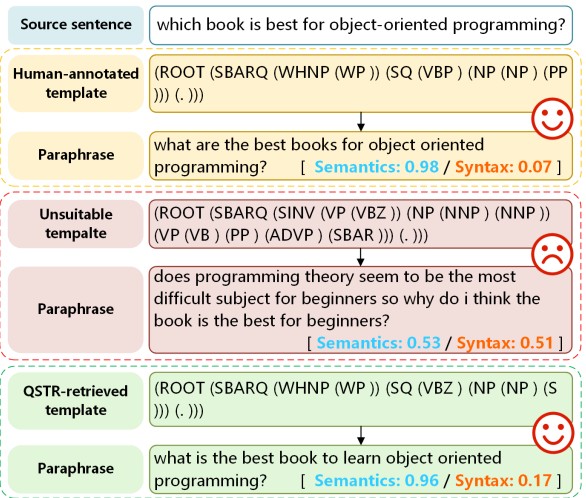

Figure 1: The generated paraphrases with different templates. "Semantics" and "Syntax" represent the semantic similarity with the source sentence and the syntactic distances against the template, respectively. Obviously, an unsuitable template may lead to a poor paraphrase.

generation (SPG) is proposed and attracts extensive attention in the research community (Iyyer et al., 2018; Chen et al., 2019a; Kumar et al., 2020; Sun et al., 2021; Hosking and Lapata, 2021; Yang et al., 2021b, 2022a; Huang et al., 2022; Hosking et al., 2022). Different from traditional PG models, SPG models take syntactic templates as additional conditions to generate paraphrases conformed with the corresponding syntactic structures, whose forms generally include syntax parse trees[1] and sentence exemplars. After years of research, SPG models can already generate syntax-conforming and high-quality paraphrases with human-annotated or well-selected syntactic templates (refer to the first case in Figure 1).

However, such promising performance heavily relies on those satisfying templates, whose difficult acquisition in practice largely hinders the applica-

---

[*]Yufeng Chen is the corresponding author.

[1]In this paper, we use syntax parse trees as templates following most previous work (Sun et al., 2021; Yang et al., 2022a; Huang et al., 2022).

tion of SPG models. Firstly, manually tailoring templates for every source sentence is practically unfeasible since it is time-consuming and laborious. Alternatively, automatically retrieving decent templates is also difficult, and unsuitable templates would induce semantic deviation or syntactic errors in the generated paraphrases (refer to the second case in Figure 1). On templates retrieval, current solutions are mostly heuristic methods and assume that suitable templates should satisfy certain conditions, *e.g.*, high frequency in the corpus (Iyyer et al., 2018) or high syntactic similarity on the source side (Sun et al., 2021). The only exception is Yang et al. (2022a), which utilizes contrastive learning to train a retriever. Nonetheless, it also assumes that a suitable template should be well-aligned with the corresponding source sentence. Albeit plausible, the retrieval standards in these methods have no guarantee of the quality of the generated paraphrases, which may lead to the unstable performance of SPG models in practice.

To address this limitation, we propose a novel Quality-based Syntactic Template Retriever (QSTR) to retrieve templates that can directly improve the quality of the paraphrases generated by SPG models. Different from previous methods, given a source sentence and a candidate template, QSTR scores the template by estimating the quality of the to-be-generated paraphrase beforehand. To achieve this, we train QSTR by aligning its output score with the real quality score of the generated paraphrase based on a paraphrase-quality-based metric, *i.e.*, ParaScore (Shen et al., 2022). With sufficient alignment, templates with higher scores from QSTR would be more probable to produce high-quality paraphrases. Moreover, when generating multiple paraphrases for each source sentence, we observe a common problem in QSTR and previous methods that the top-retrieved templates tend to be similar, which may result in similar or even repeated paraphrases. Aiming at this, we further design a Diverse Templates Search (DTS) algorithm to enhance the diversity between multiple paraphrases by restricting maximum syntactic similarity between candidate templates.

Experiments on two benchmarks demonstrate that QSTR can retrieve better templates than previous methods, which help existing SPG models generate paraphrases with higher quality. Additionally, the automatic and human evaluation showcase that our DTS algorithm can significantly improve

current retrieval methods on the diversity between multiple paraphrases and meanwhile maintain their high quality. In the end, using templates from QSTR to generate paraphrases for data augmentation achieves better results than previous retrieval methods in two text classification tasks, which further indicates the potential of QSTR in downstream applications.

In summary, the major contributions of this paper are as follows[2]:

- We propose a novel Quality-based Syntactic Template Retriever, which can retrieve suitable syntactic templates to generate high-quality paraphrases.

- To reduce the repetition when retrieving multiple templates by current methods, we design a diverse templates search algorithm that can increase the mutual diversity between different paraphrases without quality loss.

- The automatic and human evaluation results demonstrate the superiority of our method, and the performance in data augmentation for downstream tasks further prove the application values of QSTR in practical scenarios.

## 2 Related Work

As the syntactically-controlled paraphrase generation task is proposed (Iyyer et al., 2018) and has received increasing attention, previous work mainly focuses on improving the performance of the generated paraphrases conforming to the corresponding human-annotated templates. More specifically, most of them modify the model structures to better leverage the syntactic information of templates (Kumar et al., 2020; Yang et al., 2021a, 2022a). Furthermore, Sun et al. (2021) and Yang et al. (2022b) generate paraphrases based on the pre-trained language models, *e.g.*, BART (Lewis et al., 2020), ProphetNet (Qi et al., 2020), and yield better performance.

Compared to the concentration on SPG models, only rare methods focus on how to obtain templates for these SPG models in practice. Among them, Iyyer et al. (2018) and Huang and Chang (2021) directly use the most frequent templates in the corpus. Sun et al. (2021) select the syntax parse trees of the target sentences in the corpus

---

[2]The code is publicly available at: `https://github.com/XZhang00/QSTR`

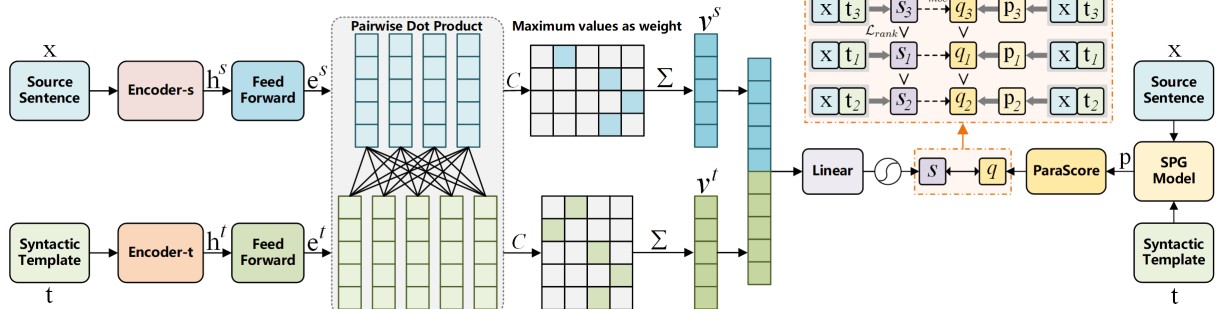

Figure 2: The model architecture and the training process of QSTR. QSTR models the relationship between the source sentence **x** and the template **t** and maps it into a score $s$, which denotes a quality estimation of the paraphrase to be generated. Then, the training objective is to align the score $s$ with the true quality score $q$ of the paraphrase **p** based on the ParaScore metric.

whose source sentences are syntactically similar to the input source sentence. Yang et al. (2022a) retrieve candidate templates based on distance in the embedding space. However, the retrieval standards in these heuristic methods cannot guarantee the quality of generated paraphrases. On the contrary, our QSTR directly predicts the quality of the paraphrases to be generated with the template, and the retrieved templates have greater potential to generate high-quality paraphrases.

## 3 Methodology

In this section, we first introduce our QSTR (§3.1), which includes the model architecture (§3.1.1) and the training objective (§3.1.2). Then we design a diverse templates search algorithm to improve the mutual diversity between multiple paraphrases (§3.2).

### 3.1 QSTR: Quality-based Syntactic Template Retriever

#### 3.1.1 Model Architecture

As shown in Figure 2, QSTR has a two-tower architecture that contains a sentence encoder and a syntactic template encoder. The two encoders are used to encode the source sentence **x** and the syntactic template **t** respectively. Formally, given a source sentence with $n$ tokens $\mathbf{x} = \{x_1, x_2, \ldots, x_n\}$ and a template[3] with $m$ constituents $\mathbf{t} = \{t_1, t_2, \ldots, t_m\}$, the sentence encoder Enc-s embeds **x** into sentence embeddings $\mathbf{h}^s$ and the syntactic template encoder Enc-t embeds **t** into template embeddings $\mathbf{h}^t$:

$$\mathbf{h}^s = (h_1^s, h_2^s, ..., h_n^s) = \text{Enc-s}(x_1, x_2, ..., x_n),$$

---

[3]For example, the template "(ROOT (S (VP (LS ) (S (VP ))) (. )))" is formalized as {'(', 'ROOT', '(', 'S', '(', ... , ')', ')'}.

$$\mathbf{h}^t = (h_1^t, h_2^t, ..., h_m^t) = \text{Enc-t}(t_1, t_2, ..., t_m).$$

To further extract the semantic and syntactic features, we add two feed-forward networks FFN-s and FFN-t after the two encoders. Then we can obtain the final sentence embeddings $\mathbf{e}^s = \{e_1^s, e_2^s, \ldots, e_n^s\}$ and the final template embeddings $\mathbf{e}^t = \{e_1^t, e_2^t, \ldots, e_m^t\}$:

$$\mathbf{e}^s = \text{FFN-s}(\mathbf{h}^s), \tag{1}$$

$$\mathbf{e}^t = \text{FFN-t}(\mathbf{h}^t). \tag{2}$$

To model the mapping relationships between the tokens in a sentence and the constituents of a related template, we calculate the pairwise dot product between two kinds of embeddings and obtain the correlation matrix $C$:

$$C_{n \times m} = \mathbf{e}^s \cdot (\mathbf{e}^t)^T, \tag{3}$$

where $C_{ij}$ indicates the degree of correlation between the token $x_i$ and the syntactic constituent $t_j$. Then we take the maximum value of each row/column in $C$ as the weight for weighted averaging sentence/template embeddings and obtain their final representations $v^s$ and $v^t$:

$$v^s = \frac{1}{n} \sum_{i=1}^{n} \left( \max_{j=1}^{m}(C_{ij}) * e_i^s \right), \tag{4}$$

$$v^t = \frac{1}{m} \sum_{j=1}^{m} \left( \max_{i=1}^{n}(C_{ij}) * e_j^t \right). \tag{5}$$

In the end, we concatenate $v^s$ and $v^t$ and transform it to a scalar $s$ through a linear layer and a Sigmoid function:

$$s = \text{Sigmoid}(W \cdot [v^s; v^t]), \tag{6}$$

where $s$ is a score that represents the matching degree between the source sentence and the syntactic template.

In a nutshell, given a sentence $\mathbf{x}$ and a template $\mathbf{t}$, the goal of QSTR is to output a quality estimation $s$ for the future paraphrase through modeling the interaction between $\mathbf{x}$ and $\mathbf{t}$:

$$s = \text{QSTR}(\mathbf{x}, \mathbf{t}). \quad (7)$$

### 3.1.2 Training Objective

During training, we aim to align the estimated score $s$ from QSTR with the real quality value of the paraphrase. Towards this end, for each $\mathbf{x}$, we randomly sample some templates from the whole template library $\mathbf{T}$ as the candidate template set $\mathbf{T}_k$, which also includes the templates of $\mathbf{x}$ and the reference $\mathbf{y}$ as more competitive candidates. Then, given $\mathbf{T}_k = \{\mathbf{t}_1, \mathbf{t}_2, \ldots, \mathbf{t}_k\}$, we can obtain the prior estimations $\mathbf{S}_k = \{s_1, s_2, \ldots, s_k\}$ for these templates from QSTR by Eq.(7). At the same time, we can use the SPG model to generate the corresponding paraphrases $\mathbf{P}_k = \{\mathbf{p}_1, \mathbf{p}_2, \ldots, \mathbf{p}_k\}$ based on $\mathbf{T}_k$ and evaluate their real quality. To acquire the quality scores of these paraphrases, we select the reference-based metric ParaScore$_{ref}$ (Shen et al., 2022) that has the highest correlation with human evaluation. Formally, given a source sentence $\mathbf{x}$ and its reference $\mathbf{y}$, the quality value $q_i$ of the paraphrase $\mathbf{p}_i$ can be calculated by:

$$q_i = \text{ParaScore}_{ref}(\mathbf{p}_i, \mathbf{x}, \mathbf{y}), \quad (8)$$

where we can use $q_i$ to construct the quality set $\mathbf{Q}_k = \{q_1, q_2, \cdots, q_k\}$.

Next, we use the Mean Square Error (MSE) loss to quickly align the prior predictions $\mathbf{S}_k$ with the posterior quality $\mathbf{Q}_k$ quantitatively:

$$\mathcal{L}_{mse} = \text{MSE}(\mathbf{S}_k, \mathbf{Q}_k). \quad (9)$$

Moreover, to better learn the quality ranks among $\mathbf{T}_k$, we also calculate a pairwise rank loss $\mathcal{L}_{rank}$ for $\mathbf{S}_k$ according to $\mathbf{Q}_k$:

$$\mathcal{L}_{rank} = \sum_{i<j} \max\left((\delta_{ij}^s - \delta_{ij}^q) * \mathbb{1}_{\delta_{ij}^q<0}, 0\right), \quad (10)$$

where $\delta_{ij}^s = s_i - s_j$, $\delta_{ij}^q = q_i - q_j$, and $\mathbb{1}_{\delta_{ij}^q<0} = 1$ when $\delta_{ij}^q < 0$ otherwise 0.

In the end, the overall training objective $\mathcal{L}$ consists of the above two loss functions:

$$\mathcal{L} = \lambda_1 \mathcal{L}_{mse} + \lambda_2 \mathcal{L}_{rank}. \quad (11)$$

---

**Algorithm 1** Diverse Templates Search Algorithm

---

**Input:** input sentence $\mathbf{x}$, the whole template library $\mathbf{T} = \{\mathbf{t}_1, \mathbf{t}_2, \ldots, \mathbf{t}_{|\mathbf{T}|}\}$, the number of the retrieved templates $d$, the syntactic diversity threshold $\beta$

**Output:** diverse templates set $\mathbf{T}_d$

1: Initialize $\mathbf{T}_d = \varnothing$ as a min heap.
2: **for** each $i \in [1, |\mathbf{T}|]$ **do**
3:     $s_i = \text{QSTR}(\mathbf{x}, \mathbf{t}_i)$
4:     **if** $|\mathbf{T}_d| < d$ **then**
5:         $\mathbf{T}_d.\text{push}(\mathbf{t}_i)$
6:     **else if** $\min \text{TED}(\mathbf{t}_i, \mathbf{T}_d[*]) > \beta$ and $s_i > s_{\mathbf{T}_d[0]}$ **then**
7:         $\mathbf{T}_d.\text{pop}(\mathbf{T}_d[0])$
8:         $\mathbf{T}_d.\text{push}(\mathbf{t}_i)$
9:     **else**
10:        continue
11:     **end if**
12:     $\mathbf{T}_d.\text{heap\_sort}()$ by $s_{\mathbf{T}_d[*]}$
13: **end for**
14: **return** $\mathbf{T}_d$

---

### 3.2 Diverse Templates Search Algorithm

In practice, the top templates retrieved by existing retrieval methods may have similar features, *e.g.*, syntactic structures, which may lead to repetitions when generating multiple paraphrases for one source sentence. To improve the mutual diversity between multiple paraphrases while maintaining their high quality, we design a general diverse templates search (DTS) algorithm as described in Algorithm 1, which can be equipped with existing retrieval methods.

Taking QSTR as an example, given an input sentence $\mathbf{x}$, we traverse the whole template library $\mathbf{T} = \{\mathbf{t}_1, \mathbf{t}_2, \ldots, \mathbf{t}_{|\mathbf{T}|}\}$ and calculate a score $s_i$ for each $\mathbf{t}_i$ (lines 2~3). Then, we maintain a min heap $\mathbf{T}_d$ in a size $d$ to collect the satisfactory templates $\mathbf{t}_i$ from $\mathbf{T}$, which have high scores and meanwhile diverse syntactic structures between each other (lines 4~12). To find these templates, we calculate the Tree Edit Distance (TED) (Zhang and Shasha, 1989) between the template $\mathbf{t}_i$ and templates in $\mathbf{T}_d$ and ensure that the minimum TED value is greater than a threshold $\beta$ before appending $\mathbf{t}_i$ to $\mathbf{T}_d$ (line 6). After one traversal, the heap $\mathbf{T}_d$ will contain the final $d$ qualified templates for diversely rephrasing the source sentence $\mathbf{x}$.

## 4 Experimental Settings

### 4.1 Datasets and Implementation Details

**Datasets.** Following previous work (Sun et al., 2021; Yang et al., 2022a), we conduct our experiments on ParaNMT-Small (Chen et al., 2019b) and QQP-Pos (Kumar et al., 2020). Specifically, ParaNMT-Small contains about 500K paraphrase pairs for training, and 1300 manually labeled (source sentence, exemplar sentence, reference sentence) triples, which are split into 800/500 for the test/dev set. And QQP-Pos contains about 140K/3K/3K pairs/triples/triples for the train/test/dev set. In the test/dev set of two datasets, the exemplars are human-annotated sentences with similar syntactic structures as the reference sentences but different semantics. The function of exemplars is to provide their syntax parse trees as templates that guide SPG models to generate paraphrases syntactically close to references.

**SPG Models.** In our experiments, we use two strong SPG models, *i.e.*, AESOP (Sun et al., 2021) and SI-SCP (Yang et al., 2022a), to evaluate the effectiveness of our QSTR. To the best of our knowledge, AESOP[4] has state-of-the-art performance among previous SPG models. Besides, SI-SCP[5] includes a novel tree transformer to model parent-child and sibling relations in the syntax parse trees and also achieves competitive performance.

**Implementation Details.** We use the Stanford CoreNLP toolkit[6] (Manning et al., 2014) to obtain the syntax parse trees of the sentences, and we truncate all parse trees by height 4 and linearise them following Yang et al. (2022a). Then, we build the template library using the parse trees from the training set of each dataset. The two encoders in QSTR are initialized with Roberta-base (Liu et al., 2019) for both datasets. We use the scheduled AdamW optimizer with a learning rate of 3e-5 during training. The max length of sentences and templates are 64 and 192 respectively. The batch size is set to 32, and the size $k$ of $T_k$ is set to 10. We train our QSTR for 10 and 20 epochs on ParaNMT-Small and QQP-Pos respectively. The coefficients $\lambda_1$ and $\lambda_2$ in Eq.(11) are all set to 1. And the threshold $\beta$ in the DTS algorithm is set to 0.2.

### 4.2 Contrast Methods

Here we introduce all the contrast methods for obtaining templates. Moreover, we also conduct paraphrase generation with Vicuna-13B[7] (Chiang et al., 2023) as a strong baseline.

**Ref-as-Template.** The syntax parse tree of the reference sentence is used as the template, which can be regarded as the ideal template for the source sentence.

**Exemplar-as-Template.** The syntax parse tree of the exemplar sentence is used as the template, which can be seen as the human-annotated template.

**Random Template.** We randomly select one template from the template library for SPG.

**Freq-R.** Following Iyyer et al. (2018), we choose the most frequent template in the template library for SPG.

**AESOP-R.** Sun et al. (2021) select the parse tree of the target sentence whose corresponding source sentence has the most similar syntactic structure to the input sentence.

**SISCP-R.** Yang et al. (2022a) encode the sentences and the templates into the same space and retrieve syntactic templates based on the similarities between their representations.

**Vicuna-13B.** We use this large language model (LLM) (Chiang et al., 2023) for zero-shot paraphrase generation. Please refer to Appendix A for more details.

### 4.3 Evaluation Metrics

**Semantic Metrics.** We use BLEU-R (Papineni et al., 2002) to evaluate literal similarity between generated paraphrases and references. To further measure the semantic similarity, we use sentence transformer[8] to encode the sentences into embeddings and then calculate the cosine similarity between the paraphrase and the source/reference sentence as cos-S/cos-R.

**Syntactic Metric.** Following Bandel et al. (2022), we calculate the Tree Edit Distance (TED) (Zhang and Shasha, 1989) between the syntax trees of the paraphrase and the template to reflect how

---

[4] https://github.com/PlusLabNLP/AESOP
[5] https://github.com/lanse-sir/SI-SCP
[6] https://stanfordnlp.github.io/CoreNLP/

[7] https://lmsys.org/blog/2023-03-30-vicuna/
[8] https://huggingface.co/sentence-transformers/all-mpnet-base-v2

| Templates | BLEU-S↓ | BLEU-R↑ | iBLEU↑ | cos-S↑ | cos-R↑ | ParaScore$_{free}$↑ | ParaScore$_{ref}$↑ | TED↓ |
|---|---|---|---|---|---|---|---|---|
| **QQP-Pos** | | | | | | | | |
| *Ideal Templates* | | | | | | | | |
| Ref-as-Template | 22.430 | 60.260 | 43.722 | 0.850 | 0.914 | 0.856 | 0.948 | 0.096 |
| Exemplar-as-Template | 21.060 | 47.270 | 33.604 | 0.836 | 0.881 | 0.848 | 0.909 | 0.132 |
| *Available Templates in Practice* | | | | | | | | |
| Random Template | **15.030** | 10.480 | 5.378 | 0.728 | 0.703 | 0.794 | 0.739 | 0.214 |
| Freq-R (Iyyer et al., 2018) | 17.210 | 16.050 | 9.398 | 0.814 | 0.789 | 0.821 | 0.787 | 0.193 |
| AESOP-R (Sun et al., 2021) | 24.430 | 13.230 | 5.698 | 0.789 | 0.742 | 0.813 | 0.769 | 0.236 |
| SISCP-R (Yang et al., 2022a) | 28.180 | 18.580 | 9.228 | **0.851** | 0.809 | 0.837 | 0.825 | 0.221 |
| **QSTR (ours)** | 20.260* | **22.080*** | **13.612** | 0.834 | **0.820*** | **0.851*** | **0.828*** | **0.163*** |
| *LLM-based paraphrase generation* | | | | | | | | |
| Vicuna-13B (Chiang et al., 2023) | 13.302 | 5.060 | 1.388 | 0.851 | 0.762 | 0.861 | 0.786 | / |
| **ParaNMT-small** | | | | | | | | |
| *Ideal Templates* | | | | | | | | |
| Ref-as-Template | 8.630 | 36.710 | 27.642 | 0.728 | 0.809 | 0.877 | 0.882 | 0.107 |
| Exemplar-as-Template | 7.000 | 22.990 | 16.992 | 0.676 | 0.719 | 0.844 | 0.805 | 0.144 |
| *Available Templates in Practice* | | | | | | | | |
| Random Template | **6.160** | 7.700 | 4.928 | 0.569 | 0.549 | 0.761 | 0.629 | 0.180 |
| Freq-R (Iyyer et al., 2018) | 8.740 | 12.510 | 8.260 | 0.671 | 0.653 | 0.834 | 0.726 | 0.226 |
| AESOP-R (Sun et al., 2021) | 7.970 | 8.770 | 5.422 | 0.649 | 0.613 | 0.816 | 0.691 | 0.185 |
| SISCP-R (Yang et al., 2022a) | 7.960 | 10.570 | 6.864 | 0.675 | 0.643 | 0.837 | 0.749 | 0.147 |
| **QSTR (ours)** | 7.530* | **13.970*** | **9.670** | **0.690*** | **0.685*** | **0.860*** | **0.769*** | **0.123*** |
| *LLM-based paraphrase generation* | | | | | | | | |
| Vicuna-13B (Chiang et al., 2023) | 10.564 | 5.279 | 2.110 | 0.731 | 0.666 | 0.861 | 0.724 | / |

Table 1: Performance of paraphrases generated with different kinds of templates using the AESOP model for SPG. Metrics with ↑ mean the higher value is better, while ↓ means the lower value is better. Results highlighted in **bold** and underline represent the best and the second-best results respectively. And results with mark * are statistically better than the most competitive method "SISCP-R" with $p < 0.05$. For all retrieval methods in "*Available Templates in Practice*", we use the top-1 retrieved template for each source sentence to generate the paraphrase.

much the paraphrase is syntactically conformed with the template. In our experiments, we observe that lower TED values generally indicate that the templates are more suitable for the source sentence.

**Diversity Metric.** BLEU-S calculates the BLEU scores between the paraphrase and the source sentence, whose lower values generally represent the paraphrases are more literally diverse from the source sentences.

**Comprehensive Metrics.** Based on BLEU-S and BLEU-R, we calculate the iBLEU (Sun and Zhou, 2012) to measure the overall quality of paraphrases by iBLEU $= \alpha$BLEU-R$ - (1-\alpha)$BLEU-S, where we set $\alpha = 0.8$ following Hosking et al. (2022). Additionally, ParaScore (Shen et al., 2022) is the state-of-the-art metric for paraphrase quality evaluation, which can comprehensively evaluate semantic consistency and expression diversity of paraphrases. Therefore, we also use the reference-based and reference-free versions of ParaScore, *i.e.*, ParaScore$_{ref}$ and ParaScore$_{free}$, as more convincing metrics. Between them, ParaScore$_{free}$ can better reflect the quality of paraphrases when reference sentences are unknown in practical scenarios.

# 5 Results

## 5.1 Main Results

Table 1 shows the performance of paraphrases with different templates on two datasets, using the AESOP model for SPG. The results based on the SISCP model are presented in Appendix B. Totally, several conclusions can be drawn from the results:

(1) Under the reference-based metrics, *i.e.*, BLEU-R, cos-R, ParaScore$_{ref}$ and iBLEU, our QSTR significantly surpasses other baselines, which demonstrates that the templates retrieved by QSTR are closer to the ideal templates.

(2) The results of the reference-free metrics (*i.e.*, BLEU-S, cos-S and ParaScore$_{free}$) also verify the superiority of our QSTR compared to other methods in practical scenarios and QSTR perform fully comparably with the human-labeled exemplar sentences (Exemplar-as-Template).

(3) QSTR also achieves a much lower TED value than other retrieval methods, which indicates that the templates from QSTR are more suitable for the source sentence and the generated paraphrases conform more with the templates.

| Templates | Rep Rate (%)↓ | M-BLEU ↓ | ParaScore$_{free}$ ↑ |
|---|---|---|---|
| SISCP-R | 14.01 | 30.70 | 0.840 |
| + DTS (ours) | 2.70 | 22.73 | 0.839 |
| QSTR (ours) | 15.48 | 32.90 | 0.857 |
| + DTS (ours) | 5.64 | 26.58 | 0.856 |

Table 2: The evaluation results of the paraphrases generated with the top-10 retrieval templates for each source sentence on the QQP-Pos dataset.

| Training Objectives | PCC (%) in Dev | PCC (%) in Test |
|---|---|---|
| QSTR | 57.51 | 57.46 |
| *w/o* $\mathcal{L}_{mse}$ | 56.70 (0.81↓) | 56.78 (0.68↓) |
| *w/o* $\mathcal{L}_{rank}$ | 56.53 (0.98↓) | 56.48 (0.98↓) |

Table 3: Pearson correlation coefficients (PCC) between QSTR predictions and ParaScore$_{ref}$ scores under different training objectives on the dev/test set of the QQP-Pos dataset.

| Retrieval Methods | Quality ↑ | Diversity↑ | Acceptance Rate (%)↑ |
|---|---|---|---|
| SISCP-R | 3.689 | 3.472 | 43.07 |
| QSTR (ours) | 3.837 | 3.752 | 53.47 |
| QSTR + DTS (ours) | **4.188** | **3.988** | **70.67** |

Table 4: The results of human evaluation.

Although some methods can achieve lower BLEU-S scores than QSTR (*e.g.*, "Random Templates"), the corresponding cos-S scores are also significantly inferior, which means the generated paraphrases with these templates have poor semantic consistency with the source sentences.

Furthermore, despite the unfair comparison, we also report the results of Vicuna-13B, which conducts zero-shot paraphrasing without the need for templates. Although using a much smaller SPG model, QSTR can yield the closest performance to Vicuna-13B among template retrieval methods. And the detailed analysis is shown in Appendix A. In conclusion, these results sufficiently demonstrate that our QSTR can provide more suitable templates to generate high-quality paraphrases.

## 5.2 Mutual Diversity of Multiple Paraphrases

In this section, we evaluate the effectiveness of our DTS algorithm when generating multiple paraphrases for each source sentence. On the evaluation metrics, we first calculate the sentence-level repetition rate (Rep-Rate) between the paraphrases generated with the top-10 retrieval templates. And we use $mutual$-BLEU (M-BLEU) to measure the literal similarity between multiple paraphrases, which averages the corpus-level BLEU scores between different paraphrases. Additionally, we also report the average ParaScore$_{free}$ scores of 10 paraphrases for quality evaluation.

Table 2 presents the results of our DTS algorithm on two retrieval methods when retrieving 10 templates. The results showcase that equipped with DTS, the values of Rep-Rate and M-BLEU are decreased significantly, which means the DTS algorithm can effectively improve the mutual diversity between multiple paraphrases. Moreover, the stable ParaScore$_{free}$ scores of these paraphrases prove that the DTS algorithm has little impact on the quality of the paraphrases. To sum up, by combining our QSTR with the DTS algorithm, the SPG models can generate multiple paraphrases with both high mutual diversity and high quality.

## 6 Analysis

### 6.1 Ablation Study

To verify the effectiveness of the two training objectives for QSTR, we further conduct an ablation study. Specifically, we calculate the Pearson correlation coefficient (PCC) between the predicted scores from QSTR and the true quality scores from ParaScore$_{ref}$. Table 3 presents the results of QSTR when $\mathcal{L}_{mse}$ or $\mathcal{L}_{rank}$ is removed during training, which show that the correlations decline obviously without $\mathcal{L}_{rank}$ or $\mathcal{L}_{mse}$. Thus, both objectives benefit the training of QSTR and they can be complementary to each other.

### 6.2 Human Evaluation

We further conduct the human evaluation on the paraphrases from the three retrieval methods (SISCP-R, QSTR, QSTR+DTS). Specifically, we randomly select 50 source sentences from the QQP-Pos test set and generate 5 paraphrases for each sentence using the AESOP model with templates from the three retrieval methods. Next, we let three annotators score each paraphrase from two aspects, *i.e.*, the overall quality (1∼5) and the diversity against the source sentence (1∼5). The detailed guidelines are listed in Appendix C. Besides, we also define a paraphrase can be accepted if its quality score ≥ 4 and diversity score ≥ 3 and it is unique among the 5 paraphrases. The final evaluation results in Table 4 show that the paraphrases generated with QSTR have better quality and diversity under human evaluation. Moreover, the DTS can further promote the quality and diversity of the paraphrases and largely improve the acceptance rate.

| | source sentence: which book would you recommend to improve english ? | | |
|---|---|---|---|
| | **Random Templates** | | **SISCP-R** |
| 1 | why did n't english improve ? | 1 | which are the best books to improve english ? i wan na ask someone please |
| 2 | is it really too late to learn english ? | 2 | what books would you recommend me to improve my english ? |
| 3 | how is english good for learning ? | 3 | what books would you recommend ? |
| 4 | what is best english book or book ? | 4 | what books would you recommend english ? |
| 5 | what are good english reading or learning book ? | 5 | which are the best english books ? |
| 6 | which book would be the best to improve english ? | 6 | which books are the best for me to improve my english ? |
| 7 | how do i actually learn english ? | 7 | what book would you recommend improve english ? |
| 8 | can anyone help me improve english ? | 8 | what book would you recommend me to improve my english ? |
| 9 | how has one improved his english ? | 9 | what books should i read to improve my english ? |
| 10 | what would you suggest or books to improve english ? | 10 | which book should i read to improve my english ? |
| | **QSTR (ours)** | | **QSTR + DTS (ours)** |
| 1 | what is the best book for improving english ? | 1 | what is the best book for improving english ? |
| 2 | which is the best book to improve english ? | 2 | which are the best books to improve english ? |
| 3 | what are some of the best books for improving english ? | 3 | how do i improve my spoken english ? |
| 4 | what is the best book to improve my spoken english soon ? | 4 | which english books you would recommend to improve english ? |
| 5 | what are some good books for improving english ? | 5 | what books should i read to improve my spoken english ? |
| 6 | what are some good books or resources to improve english ? | 6 | which english book to buy to improve my spoken english ? |
| 7 | which are the best books to improve english ? | 7 | which one is the best book to improve english ? |
| 8 | which is the best book for improving english ? | 8 | what english books you would recommend to improve your pronunciation ? |
| 9 | what are the best books for improving english ? | 9 | which english book should i read ? |
| 10 | what are the best books to improve english ? | 10 | which books or magazines would you recommend me to improve my english ? |

Table 5: Paraphrases generated by the AESOP model with templates retrieved by different methods. The paraphrases in red/blue/black represent that they are terrible/repeated/eligible.

| Retrieval Methods | Train Aug | | | Test Aug | Train & Test Aug |
|---|---|---|---|---|---|
| | MRPC | QQP | SST-2 | SST-2 | SST-2 |
| Few-shot baseline | 70.750 | 70.500 | 86.546$^\dagger$ | 86.546$^\dagger$ | 87.864$^\ddagger$ |
| + Random templates | 70.333 | 69.833 | 86.216 | 86.875 | 88.029 |
| + Freq | 71.833 | 71.167 | 85.832 | 86.985 | 87.974 |
| + AESOP-R | 71.500 | 72.000 | 87.040 | 87.095 | 88.138 |
| + SISCP-R | 71.583 | 71.917 | 86.930 | 87.150 | 88.468 |
| + QSTR (ours) | **72.250** | **72.417** | **87.864$^\ddagger$** | **87.534** | **88.523** |

Table 6: Test accuracies of downstream tasks (*i.e.*, MRPC, QQP, and SST-2) after adding paraphrases with different templates respectively to the original baseline for data augmentation. "Train Aug" means generating paraphrases for the training samples as the training corpus. "Test Aug" represents generating paraphrases for the test samples and conducting majority voting for the final predictions. "Train & Test Aug" combines the aforementioned two strategies. And results with the same mark $^\dagger$ or $^\ddagger$ are from the same model.

## 6.3 Case Study

We list 10 generated paraphrases of different retrieval methods for the same source sentence in Table 5. Among them, random templates produce the most inferior paraphrases, which shows that current SPG models are very sensitive to different templates. With the templates retrieved by SISCP-R, the paraphrases may be similar to each other and also have some syntactic or semantic errors. In contrast, our QSTR performs better on the quality of the paraphrases and the DTS algorithm further improves the mutual diversity of paraphrases.

## 6.4 Applications on Downstream Tasks

To further test the performance of QSTR on downstream tasks, we apply it to augment data for few-shot learning in text classification tasks. Specifically, we select SST-2, MRPC, and QQP classification tasks from GLUE (Wang et al., 2018) as evaluation benchmarks. Then, we randomly sample 200 instances from the train set to fine-tune `bert-base-uncased` (Devlin et al., 2019) to obtain the baseline classifier as the few-shot baseline. In addition, we utilize the AESOP model with templates from different retrieval methods to generate paraphrases for the train set and the test set respectively. Specifically, the augmented data for the train set are used to train classifiers together with the original instances. As for the test set, we evaluate the augmented data as additional results, getting the majority voting as the final results. Moreover, we combine the aforementioned two strategies as a further attempt. Please refer to Appendix D for more training details.

The results in Table 6 present that our method brings the highest improvement over the baseline

compared to other methods on the three strategies. Specifically, "Train Aug" leads to better performance with our QSTR but not stably with other methods. "Test Aug" contributes to stable improvements with all methods. And "Train & Test Aug" further improves the final performance. In conclusion, our QSTR showcases the best performance under all strategies, which indicates that our method can effectively promote the application values of SPG models on downstream tasks.

## 7 Conclusion

In this work, we propose a quality-based template retriever (QSTR) to retrieve decent templates for high-quality SPG. Moreover, we develop a diverse templates search (DTS) algorithm to reduce the repetitions in multiple paraphrases. Experiments show that the SPG models can generate better paraphrases with the templates retrieved by our QSTR than other retrieval methods and our DTS algorithm further increases the mutual diversity of the multiple paraphrases without any loss of quality. Furthermore, the results of the human evaluation and the downstream task also demonstrate that our QSTR and DTS algorithm can retrieve better templates and help SPG models perform more stably in practice.

## Limitations

Although Parascore$_{ref}$ (Shen et al., 2022)) has been the state-of-the-art metric for the quality evaluation of paraphrases, it is still far from perfect as the supervision signal for QSTR. We will explore better metrics for evaluating the quality of paraphrases to guide the training of QSTR in future work. Moreover, we only utilize Vicuna-13B for zero-shot paraphrase generation, which leads to an unfair comparison with other methods. In future work, We will try to finetune Vicuna-13B on the SPG task and verify the effectiveness of our method with this new backbone.

## Acknowledgements

The research work described in this paper has been supported by the National Natural Science Foundation of China (NSFC) via grants 61976016 and 61976015 and the Talent Fund of Beijing Jiaotong University via grant 2023XKRC006. The authors would like to thank the anonymous reviewers for their valuable comments and suggestions to improve this paper.

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

## A   The experiment based on Vicuna-13B

Vicuna-13B (Chiang et al., 2023) is a large-scale model that trained by fine-tuning LLaMA on user-shared conversations collected from ShareGPT. And preliminary evaluation using GPT-4 as a judge shows Vicuna-13B outperforms other models like LLaMA and Stanford Alpaca in more than 90% cases (Chiang et al., 2023). To explore its performance on paraphrase generation, We use the instruction "Please give ten paraphrases of the next sentence in English, *the input sentence*. These paraphrases should have the same meaning and diverse syntactic structures with the given sentence." to obtain the generated paraphrases from Vicuna-13B. The evaluation results on automatic metrics (as shown in Table 1) represent that Vicuna-13B can generate competitive paraphrases with our QSTR. Additionally, through our observation of specific cases, Vicuna-13B tends to add more additional information or expands the original sentence length during generating paraphrases. Several cases generated from Vicuna-13B are shown in Table 7. The advantage of this performance is that the generated paraphrases are more diverse against the source sentence, while the disadvantages are that the sentences become more redundant and the syntactic structures are not uncontrollable.

## B   The Results of SI-SCP

Firstly, to show the applicability of QSTR, the QSTR model used for the SI-SCP model is the same one used for the AESOP model (QSTR-based-on-AESOP). It is trained using the paraphrases generated by the AESOP model, which may leave a gap from the paraphrases from the SI-SCP model. Thus, we also conduct the more effective approach, using the SI-SCP model to generate paraphrases during the training process (QSTR-based-on-SI-SCP). Table 8 shows the performance of paraphrases with different templates from QSTR and other baselines on both datasets based on the SI-SCP backbone. The results showcase that our QSTR exhibits significant superiority compared to other baselines, and the QSTR-based-on-SI-SCP has better performance on most metrics than QSTR-based-on-AESOP. However, the promising results with QSTR-based-on-AESOP indicate that our QSTR may be directly applied to other SPG models without retraining.

| | source sentence: why do some people like cats more than dogs ? |
|---|---|
| 1 | What is the reason behind the preference of some individuals for cats over dogs? |
| 2 | What makes certain people have a greater fondness for cats compared to dogs? |
| 3 | Why are some people more inclined towards cats rather than dogs? |
| 4 | What drives some individuals to prefer cats over dogs? |
| 5 | What is the explanation for the fact that some people prefer cats to dogs? |
| 6 | Why do certain people have a stronger affinity for cats compared to dogs? |
| 7 | What is the reason for the disparity in preferences between people who prefer cats and those who prefer dogs? |
| 8 | What sets apart the individuals who prefer cats from those who prefer dogs? |
| 9 | Why do some people find cats more appealing than dogs? |
| 10 | What are the factors that contribute to the variance in preferences for cats and dogs among people? |
| | source sentence: how can i send a private message through quora ? |
| 1 | How do I privately message someone on Quora? |
| 2 | Can you tell me the process for sending a private message on Quora? |
| 3 | I'm trying to send a confidential message on Quora, but I'm not sure how How can you help? |
| 4 | Is there a way to send a discreet message on Quora? If so, how would I go about doing it? |
| 5 | How do I go about sending a secure message on Quora? |
| 6 | I'm trying to send a message on Quora that only the recipient can see How can I do that? |
| 7 | Can you explain the steps to send a private message on Quora? |
| 8 | I'm trying to send a message on Quora that won't show up in someone's feed How can I do that? |
| 9 | How do I send a message on Quora that won't be visible to anyone else? |
| 10 | Can you provide me with the procedures to send a private message on Quora? |

Table 7: Paraphrases directly generated by Vicuna-13B.

## C Guidelines for human evaluation

The overall quality evaluates paraphrases from the perspectives of grammar correctness and semantic consistency with the source sentence, and the larger the score, the higher the quality. The detailed guidelines are as follows.

- "5" means the paraphrase is fully grammatically correct and completely semantically consistent with the source sentence.

- "4" means the paraphrase has a slight grammatical error, but still maintains the correct semantics, and can also be considered a valuable paraphrase.

- "3" means the paraphrase has a slight grammatical error and a minor semantic deviation.

- "2" means the paraphrase has a serious grammatical error and a major semantic deviation, but is still a complete sentence.

- "1" means the paraphrase is not a complete sentence or is totally irrelevant to the source sentence.

The diversity evaluates whether the paraphrase has diverse expressions compared to the input sentence based on whether the words used are the same or whether the syntactic structure is the same. And the larger the score, the higher the diversity. The detailed guidelines are as follows.

- "5" means the paraphrase has a very different syntax from the source sentence and adopts many new words and phrases.

- "4" means the paraphrase has a new syntax against the source sentence and revises some original words.

- "3" means the paraphrase has a new syntax against the source sentence but still adopts original words or phrases.

- "2" means the paraphrase has the same expression as the source sentence but adopts a few new words or phrases.

- "1" means the paraphrase is almost identical to the source sentence.

## D Training details of Downstream Tasks

Since MRPC does not provide the official dev set, we randomly sample 1200 instances from the official test set as the final testing set and use the rest instances as the dev set. QQP adopts the same settings. As for the SST-2, we directly use the

| Templates | BLEU-S↓ | BLEU-R↑ | iBLEU↑ | cos-S↑ | cos-R↑ | ParaScore$_{free}$↑ | ParaScore$_{ref}$↑ | TED↓ |
|---|---|---|---|---|---|---|---|---|
| **QQP-Pos** | | | | | | | | |
| *Ideal Templates* | | | | | | | | |
|   Ref-as-Template | 23.810 | 52.760 | 37.446 | 0.838 | 0.888 | 0.844 | 0.920 | 0.136 |
|   Exemplar-as-Template | 23.160 | 42.730 | 29.552 | 0.829 | 0.860 | 0.840 | 0.888 | 0.181 |
| *Available Templates in Practice* | | | | | | | | |
|   Random Template | **20.540** | 15.920 | 8.628 | 0.755 | 0.732 | 0.802 | 0.768 | 0.301 |
|   Freq-R (Iyyer et al., 2018) | 22.120 | 18.190 | 10.128 | 0.795 | 0.776 | 0.811 | 0.790 | **0.208** |
|   AESOP-R (Sun et al., 2021) | 28.040 | 17.190 | 8.144 | 0.798 | 0.761 | 0.806 | 0.790 | 0.323 |
|   SISCP-R (Yang et al., 2022a) | 29.230 | 21.050 | 10.994 | **0.839** | 0.805 | 0.820 | 0.817 | 0.284 |
|   **QSTR-based-on-AESOP (ours)** | 21.520 | **22.540** | **13.728** | 0.815 | 0.804 | 0.833 | 0.820 | 0.220 |
|   **QSTR-based-on-SI-SCP (ours)** | 24.580 | 22.500 | 13.084 | 0.828 | **0.810** | **0.850** | **0.829** | 0.214 |
| **ParaNMT-small** | | | | | | | | |
| *Ideal Templates* | | | | | | | | |
|   Ref-as-Template | 12.060 | 26.130 | 18.492 | 0.711 | 0.758 | 0.837 | 0.839 | 0.158 |
|   Exemplar-as-Template | 11.980 | 18.950 | 12.764 | 0.665 | 0.686 | 0.811 | 0.785 | 0.191 |
| *Available Templates in Practice* | | | | | | | | |
|   Random Template | **10.110** | 6.870 | 3.474 | 0.596 | 0.562 | 0.748 | 0.653 | 0.256 |
|   Freq-R (Iyyer et al., 2018) | 14.030 | 9.520 | 4.810 | 0.643 | 0.607 | 0.786 | 0.700 | 0.312 |
|   AESOP-R (Sun et al., 2021) | 13.340 | 9.470 | 4.908 | 0.664 | 0.615 | 0.793 | 0.711 | 0.231 |
|   SISCP-R (Yang et al., 2022a) | 17.200 | 11.810 | 6.008 | **0.723** | 0.676 | 0.807 | 0.779 | 0.190 |
|   **QSTR-based-on-AESOP (ours)** | 13.160 | **13.830** | **8.432** | 0.694 | 0.675 | 0.823 | 0.768 | **0.157** |
|   **QSTR-based-on-SI-SCP (ours)** | 16.070 | 12.770 | 7.002 | 0.710 | **0.680** | **0.833** | **0.781** | 0.167 |

Table 8: Performance of paraphrases with different kinds of templates based on the SI-SCP backbone. Metrics with ↑ means the higher value is better, while ↓ means the lower value is better. And the **bold** and underline represent the best and the second best respectively. For all retrieval methods in "*Available Templates in Practice*", we use the top-1 retrieved template for each source sentence to generate the paraphrase.

official test set and dev set. All classifiers are initialized with `bert-base-uncased` (Devlin et al., 2019). The batch size is 128, 64, and 64 for MRPC, QQP, and SST-2 respectively. The learning rate is $10^{-4}$ and the number of training epochs is 20. And we use AdamW optimizer with weight decay being $10^{-5}$.