# OpenReview forum: "A Quality-based Syntactic Template Retriever for Syntactically-Controlled Paraphrase Generation"
_EMNLP/2023/Conference — EMNLP 2023 Main_

### Official Review · Reviewer_7mbF · 2023-08-04

**Typos Grammar Style And Presentation Improvements:** N/A
**Soundness:** 3

**Excitement:**

3: Ambivalent: It has merits (e.g., it reports state-of-the-art results, the idea is nice), but there are key weaknesses (e.g., it describes incremental work), and it can significantly benefit from another round of revision. However, I won't object to accepting it if my co-reviewers champion it.

**Missing References:**

See Reasons To Reject

**Paper Topic And Main Contributions:**

This paper focuses on the syntactically-controlled paraphrase generation task and proposed a new method QSTR and DTS algorithm. In short, this method retrieves syntax-parse-tree-based templates to improve the quality of paraphrases.

**Questions For The Authors:**

Question A: Does the retrieval step affect the training/inference time of the model?

**Reasons To Accept:**

1. It's an interesting point to consider generating diversity in the syntactically-controlled paraphrase generation task.
2.  The tables and figures are pretty clear and easy to understand. Authors provide code and dataset for reproducibility.

**Reasons To Reject:**

1. Some parts of the paper are not clearly written.  For example, one of the focuses of this paper is the template, but it does not emphasize the information about the template (readers can only infer from the figure that the template refers to the syntax tree), and the choice of template is not fully discussed, such as the part of speech sequence, truncated syntax tree (after all, using the whole syntax tree may lead to too long input sequence).
2. The task is not clearly defined. As far as I know, the controlled paraphrase generation task on these two data sets generally provides only data pairs [input sentence, output sentence] at the time of training. In the testing stage, an examplar is provided and the model is required to refer to examplar examples to paraphrase results that contain specific syntactic structures. Based on this, I do not quite understand why exemplar-as-template is used as the baseline (previous articles generally Exemplar-as-Output, see article [1][2]).
3. Baselines are seriously lacking and comparisons may be unfair, and the authors do not include such strong baselines [2][3][4] in the comparison. At the same time, from the conclusions and experimental results of these papers, the use of natural language sentences as templates is obviously superior to the use of syntactic trees, so I may not quite understand the reason for using syntactic trees as templates. In addition, the method in this paper has two RoBerta-based encoders, and the baselines compared are transformer based. It is hard to determine whether performance gains are due to the use of the PLMs or the method itself.
4. The design of manual evaluation scores is not very reasonable, the author only evaluates quality and diversity, and lacks a separate evaluation of syntactic controllability, which is undoubtedly the focus of the task.

[1] Syntax-Guided Controlled Generation of Paraphrases.

[2] Generative Pre-training for Paraphrase Generation by Representing and Predicting Spans in Exemplars

[3] GCPG: A General Framework for Controllable Paraphrase Generation

[4] HCPG: a highlighted contrastive learning framework for exemplar-guided paraphrase generation

**Reproducibility:**

3: Could reproduce the results with some difficulty. The settings of parameters are underspecified or subjectively determined; the training/evaluation data are not widely available.

**Reviewer Confidence:**

5: Positive that my evaluation is correct. I read the paper very carefully and I am very familiar with related work.

---

> ### Author Rebuttal · Authors · 2023-08-29
>
> Thanks very much for your time and effort to review our paper, but we are afraid that you may have some misunderstandings about our paper. To clarify your misunderstandings, we will provide point-by-point explanations on your comments in the following.
>
> ---
>
> > **Q1**：Some parts of the paper are not clearly written. For example, one of the focuses of this paper is the template, but it does not emphasize the information about the template (readers can only infer from the figure that the template refers to the syntax tree), and the choice of template is not fully discussed, such as the part of speech sequence, truncated syntax tree (after all, using the whole syntax tree may lead to too long input sequence).
>
> **A1**：(1) Actually, we have introduced that we use syntax parse trees as templates in this paper in Footnote 2 of Page 1. To emphasize it, we will move them to the main content following your insightful suggestions.
>
> (2) On the choice of the template, we choose the syntax parse tree in this paper as it is one of the mainstream forms of the template in previous work on SPG. Besides, we totally agree with your concern about the too-long input, and thus we truncated all parse trees by height 4 and linearised them following previous work (Yang et al, 2022) as we have discussed and listed in Lines 316-318. In addition, we also present an example in Footnote 3 of Page 3 for a better understanding of their forms.
>
> Erguang Yang et al. Learning Structural Information for Syntax-Controlled Paraphrase Generation. NAACL'22 Findings.
>
> ---
>
> > **Q2**：The task is not clearly defined. As far as I know, the controlled paraphrase generation task on these two data sets generally provides only data pairs [input sentence, output sentence] at the time of training. In the testing stage, an examplar is provided and the model is required to refer to examplar examples to paraphrase results that contain specific syntactic structures. Based on this, I do not quite understand why exemplar-as-template is used as the baseline (previous articles generally Exemplar-as-Output, see article [1][2]).
>
> **A2**：In this paper, we are not aiming at proposing a new model or method for syntactically controlled paraphrase generation, but **tackling a practical issue for existing SPG models on the lack of qualified templates in practice**. To this end, we propose QSTR in this paper, which can be plugged into any existing SPG models to retrieve templates for high-quality paraphrases.
>
> In the main experiments, to simulate practical scenarios, we assume that there is no template for sentences in the test set and we need to retrieve templates from the template library with our QSTR as well as other existing solutions for obtaining templates. Then we use the same SPG model (e.g., AESOP or SISCP) to generate paraphrases based on these templates and compare the performance of the paraphrases. Under this setting, **"exemplar-as-template"**, which means using the parse trees of the exemplar sentences as templates, **is just the simulation of the human-annotated templates in practice**. As shown in Table 1/8, the templates retrieved by our QSTR achieve fully comparable performance with these human-annotated templates on reference-free metrics.
>
> ---
>
>
> > **Q3**：Baselines are seriously lacking and comparisons may be unfair, and the authors do not include such strong baselines [2][3][4] in the comparison. At the same time, from the conclusions and experimental results of these papers, the use of natural language sentences as templates is obviously superior to the use of syntactic trees, so I may not quite understand the reason for using syntactic trees as templates. In addition, the method in this paper has two RoBerta-based encoders, and the baselines compared are transformer based. It is hard to determine whether performance gains are due to the use of the PLMs or the method itself.
>
> **A3**：As we explained above, our goal is to retrieve qualified templates for each input sentence, so our baselines are template acquisition methods rather than other paraphrase models. However, the methods you mentioned aim to improve the controllability of the paraphrases given the annotated sentence exemplars as templates, which is inconsistent with our focus. Therefore, we are not comparing our RoBerta-based templated retrieval model with current Transformer-based SPG models, but with other template retrieval methods, and testing the performance of these retrieved templates when paraphrasing using the same SPG model (*e.g.*, AESOP and SISCP).
>
> According to the experimental results, the performance gains are due to our method being able to retrieve suitable templates, which can guide SPG models to generate better paraphrases.
>
> Furthermore, as reviewer TgL2 concluded, our QSTR can be plugged into any SPG models for better performance in practice, including the ones you mentioned that use sentence exemplars as templates. And we will explore this possibility in future work.
>
> ---
>
>
> > **Q4**：The design of manual evaluation scores is not very reasonable, the author only evaluates quality and diversity, and lacks a separate evaluation of syntactic controllability, which is undoubtedly the focus of the task.
>
> **A4**：In terms of syntactic controllability, we think that compared to human evaluation, Tree Edit Distance (TED) between the syntax tree of the paraphrase and the template is a more precise metric to reflect how much the paraphrase is syntactically conformed with the template. Under this metric, our QSTR has shown substantial superiority compared to other template retrieval methods.
>
> ---
>
>
> > **Q5**：Does the retrieval step affect the training/inference time of the model?
>
> **A5**：Like any other template retrieval method we have compared in experiments, the retrieval step will introduce the time cost of inference of the SPG models, which is inevitable in order to get better templates. However, the benefit of the two-tower architecture of our QSTR, we can encode all the candidate templates beforehand and largely improve the retrieval speed during application.
>
> ---
>
> We hope our responses will address your concerns and clarify the contributions of our work. We would greatly appreciate it if you could reassess our paper after reading our responses. Thanks again for your efforts and valuable comments on our paper!

---

### Official Review · Reviewer_TgL2 · 2023-08-04

**Typos Grammar Style And Presentation Improvements:** None
**Soundness:** 4

**Excitement:**

4: Strong: This paper deepens the understanding of some phenomenon or lowers the barriers to an existing research direction.

**Missing References:**

NA

**Paper Topic And Main Contributions:**

The paper tackles the problem of selecting (syntactic) templates for syntactically controlled paraphrase generation (SPG) and proposes a quality-based retriever that ranks the quality of to-be-generated paraphrase from a source sentence given a template. The retriever model is a two-tower siamese network that scores a syntactic template for a provided source sentence. Further, the authors propose a Diverse Templates Search algorithm to ensure diversity in generated paraphrases when generating multiple paraphrases from a single source sentence. Through experiments on QQP-Pos and ParaNMT-small datasets, it is shown that augmenting existing SPG methods with the proposed retriever results in a substantial improvements in the performance on both automated and human evaluation. Further, the proposed method when used for data augmentation also helps in improving downstream task performance for paraphrase detection datasets (MRPC and QQP).

**Questions For The Authors:**

None

**Reasons To Accept:**

1. The paper tackles the often overlooked aspect of syntactically controlled paraphrase generation i.e. how to select appropriate templates, since randomly selected templates might not be compatible with a source sentence. The proposed method is intuitive and can be plugged into any SPG model to improve the performance. I think both tackling an important problem and providing a scalable solution makes the work valuable to the community, especially researchers working on paraphrase generation or related problems.

2. I also really appreciate the fact that the authors highlight as well as tackle the pitfalls of a quality based retriever approach for selecting templates i.e. when generating multiple paraphrases from a single source sentence, the model might end up selecting syntactically similar templates leading to low diversity in generated paraphrases. The proposed Diverse Template Search algorithm while simple, provides an effective way of combating this issue as evident from the experimental results that show substantial improvement in diversity.

3. The evaluation is very thorough including a wide variety of baselines (including an open source LLM), automatic evaluation metrics as well as human evaluation. This helps answer most questions the reader would have about the effectiveness of the proposed approach. Evaluation of the method when acting as a method for data-augmentation for downstream tasks is also well appreciated.

**Reasons To Reject:**

I didn't find any clear weakness in the work, except that the proposed method seems to be less effective when used with SISCP SPG model compared to the performance gains observed for AESOP.

**Reproducibility:**

4: Could mostly reproduce the results, but there may be some variation because of sample variance or minor variations in their interpretation of the protocol or method.

**Reviewer Confidence:**

4: Quite sure. I tried to check the important points carefully. It's unlikely, though conceivable, that I missed something that should affect my ratings.

---

> ### Author Rebuttal · Authors · 2023-08-29
>
> Thanks very much for your positive comments and appreciating the contributions of our work. Next, we clarify the reason why templates retrieved from QSTR seem to be less effective when using SISCP as the backbone SPG model.
>
> Actually, to show the applicability of QSTR, the QSTR model used for the SISCP model is the same one that used for the AESOP model. It is trained using the paraphrases generated by AESOP model, which may leave a gap from the paraphrases from the SISCP model. Thus, the more effective approach is using SISCP model to generate paraphrases during the training process.
>
> In the table below, we have complemented the corresponding results with SISCP as the SPG model, which showcases that the QSTR trained using paraphrases generated by SISCP has better performance on most metrics than the original one trained using the paraphrases from AESOP and still exhibits significant superiority compared with other baselines. And we will revise the corresponding results to our paper in the new version.
>
> > **QQP-Pos**
>
> | Templates |  BLEU-S↓| BLEU-R↑ | iBLEU↑ | cos-S↑ | cos-R↑ | $ParaScore_{free}$ ↑| $ParaScore_{ref}$↑ | TED↓ |
> |--------------- | :----: | :----: | :----: | :----: | :----: | :----: |  :----: | :----: |
>  | Random Templates  | **20.540**  | 15.920  | 8.628  | 0.755 |  0.732  | 0.802 |  0.768  | 0.301 |
>  | Freq-R   | 22.120  | 18.190  | 10.128  | 0.795  | 0.776 |  0.811 |  0.790  | **0.208** |
>  | AESOP-R |   28.040  | 17.190  | 8.144  | 0.798 |  0.761 |  0.806 |  0.790 |  0.323 |
>  | SISCP-R  |  29.230  | 21.050  | 10.994 |  **0.839** |  0.805  | 0.820 |  0.817  | 0.284 |
> | QSTR-based-on-AESOP (original) |21.520 	 | **22.540** 	 | **13.728** 	 | 0.815 	 | 0.804 	 | 0.833 	 | 0.820 	 | 0.220  |
> | QSTR-based-on-SISCP (new) | 24.580  | 	22.500 	 | 13.084 	 | 0.828↑ 	 | **0.810**↑ 	 | **0.850** ↑	 | **0.829**↑	 | 0.214↓  |
>
> ---
>
> > **ParaNMT-small**
>
> | Templates |  BLEU-S↓| BLEU-R↑ | iBLEU↑ | cos-S↑ | cos-R↑ | $ParaScore_{free}$ ↑| $ParaScore_{ref}$↑ | TED↓ |
> |--------------- | :----: | :----: | :----: | :----: | :----: | :----: |  :----: | :----: |
>  | Random Templates  | **10.110**  | 6.870  | 3.474  | 0.596  | 0.562  | 0.748  | 0.653  | 0.256 |
>  | Freq-R   | 14.030  | 9.520  | 4.810  | 0.643  | 0.607 |  0.786 |  0.700  | 0.312 |
>  | AESOP-R  |  13.340  | 9.470  | 4.908  | 0.664  | 0.615  | 0.793  | 0.711 |  0.231 |
>  | SISCP-R   | 17.200  | 11.810  | 6.008  | **0.723** |  0.676  | 0.807  | 0.779  | 0.190 |
> | QSTR-based-on-AESOP (original) | 13.160  | 	**13.830** 	 | **8.432** 	 | 0.694 	 | 0.675 	 | 0.823 	 | 0.768 	 | **0.157**  |
> | QSTR-based-on-SISCP (new) | 16.070  | 	12.770 	 | 7.002  | 	0.710↑  | 	**0.680**↑	 | **0.833**↑ 	 | **0.781**↑  | 0.167  |

---

### Official Review · Reviewer_h28a · 2023-08-05

**Soundness:** 4

**Excitement:**

4: Strong: This paper deepens the understanding of some phenomenon or lowers the barriers to an existing research direction.

**Paper Topic And Main Contributions:**

Prior methods in Syntactic Paraphrase generation focused on generating sentences that conform to a particular syntactic style that was given as an input to the generation models. While those works aimed at solving the task of syntax-guided paraphrase generation via strong semantic and syntactic encoders and decoders, they lacked a concrete method for choosing the target syntactic styles. The paper proposes a syntactic template retriever model to select not only a good set of high quality diverse structures for syntax-guided paraphrase generation but also highly compatible ones.

To address the issue of lack of good syntactic templates, the paper proposes a method called Quality-based Syntactic Template Retriever (QSTR). QSTR involves an architecture that calculates the interaction between source sentence and syntactic templates, to come up with a compatibility signal that is brought close to paraphrase quality scores of the output sentence using a metric called ParaScore. Two objectives namely mse and rank loss help in achieving the desired result.

Additionally the paper proposes Diverse Templates Search (DTS) algorithm that prevents mode collapse between retrieved candidates. The paper shows better results than previous heuristic based template selection mechanisms. Additionally the paper reports good results on human evaluation and a downstream classification task.

**Questions For The Authors:**

1. The current objective just takes into account the syntactic signal compatibility implicitly. Would it help to include a syntactic signal objective explicitly while training QSTR?

2. In downstream tasks, would it be better to use the generations as augmentations or generate multiple instances for test set, evaluate them, get majority voting and display the final result ?

3. What are the guidelines for human evaluation ?

**Reasons To Accept:**

1. The paper is well-written and the method is easy to follow

2. Comprehensive evaluation and Comparison against competitive baselines, including Vicuna-13B (LLM).

**Reasons To Reject:**

1. Evaluation on downstream task that is similar to original task. Using paraphrase generations from DTS to evaluate paraphrase detection is helpful but not sufficient to show that diverse paraphrases are helpful for downstream tasks. It would be better to show results on non-paraphrastic tasks like sentiment classification, or NLI.

2. Statistical significance testing missing.

**Reproducibility:**

4: Could mostly reproduce the results, but there may be some variation because of sample variance or minor variations in their interpretation of the protocol or method.

**Reviewer Confidence:**

5: Positive that my evaluation is correct. I read the paper very carefully and I am very familiar with related work.

---

> ### Author Rebuttal · Authors · 2023-08-29
>
> Thanks very much for your constructive comments and suggestions, they are exceedingly helpful for us to improve our paper. We will response to your concerns point-by-point in the following.
>
> ---
>
> > **Q1**：Evaluation on downstream task that is similar to original task. Using paraphrase generations from DTS to evaluate paraphrase detection is helpful but not sufficient to show that diverse paraphrases are helpful for downstream tasks. It would be better to show results on non-paraphrastic tasks like sentiment classification, or NLI.
>
> **A1**：We are very grateful for your suggestion, and we have conducted additional experiments on SST-2 (a sentiment classification dataset from GLUE). The results in the table below present that our methods perform better compared to other baselines, which demonstrates that our methods are helpful for downstream tasks. And we will add these experiments to our paper in the new version.
>
> | Retrieval Methods	| SST-2 (%) |
> | -------------- | :---------------: |
> | Few-shot baseline	 | 86.546 |
> | &emsp;+Random templates	 | 86.216 |
> | &emsp;+Freq-R	  | 85.832 |
> | &emsp;+AESOP-R	 | 87.040 |
> | &emsp;+SISCP-R	 | 86.930 |
> | &emsp;+QSTR (ours) | 	**87.864** |
>
> ---
>
> > **Q2**：Statistical significance testing missing.
>
> **A2**：Thanks very much for your suggestion, and we have added the statistical significance testing, which shows that our QSTR significantly surpasses other methods with $p<0.05$. We will supplement the results in the new version.
>
> ---
>
> > **Q3**：The current objective just takes into account the syntactic signal compatibility implicitly. Would it help to include a syntactic signal objective explicitly while training QSTR?
>
> **A3**：In preliminary experiments, we tried to use Tree Edit Distance (TED) between the parse trees of paraphrases and templates as the syntactic supervision signal to train QSTR. However, we failed for the following reasons.
> - For each source sentence, the TED values of the generated paraphrases tend to be similar (or equal to 0) at most times, which makes it difficult to train QSTR.
> - The process of syntax parsing is slower than calculating ParaScore by about 5 times, which largely lengthens the training time.
> - In this paper, our goal is to improve the quality of the generated paraphrases using our retrieved templates, however, we found that lower TED values may not necessarily correspond to higher-quality paraphrases. For example, using the syntax parse tree of the source sentence as the template will have a very low TED value, but it will make the generated paraphrase too similar to the source sentence.
>
> Nevertheless, inspired by your question, we will explore the possibility of combining ParaScore and TED as a new objective to retrieve more compatible templates in future work.
>
> ---
>
> > **Q4**：In downstream tasks, would it be better to use the generations as augmentations or generate multiple instances for test set, evaluate them, get majority voting and display the final result ?
>
> **A4**：We have compared two methods on the SST-2 dataset, including data augmentation for the training set and the voting strategy you mentioned for the test set. The results in the table below showcase that:
> - Data augmentation for the training set performs slightly better with our QSTR but stably with other methods.
> - Majority voting for the test set leads to stable improvement for all methods.
> - Our QSTR shows the best performance under both settings.
>
> | Retrieval Methods	| for train (%) | for test (%) |
> | -------------- | :---------------: | :---------------: |
> | Few-shot baseline	 | 86.546 | 86.546 |
> | &emsp;+Random templates	 | 86.216 | 86.875 |
> | &emsp;+Freq-R	  | 85.832 | 86.985  |
> | &emsp;+AESOP-R	 | 87.040 | 87.095 |
> | &emsp;+SISCP-R	 | 86.930 | 87.150 |
> | &emsp;+QSTR (ours) | 	**87.864** | **87.534** |
>
> ---
>
> > **Q5**：What are the guidelines for human evaluation ?
>
> **A5**：In human evaluation, we let three annotators score each paraphrase from two aspects, *i.e.*,  the overall quality (1-5) and the diversity against the source sentence (1-5).
>
> (a) The overall quality evaluates paraphrases from the perspectives of grammar correctness and semantic consistency with the source sentence, and the larger the score, the higher the quality. The detailed guidelines are as follows.
>
> - "5" means the paraphrase is fully grammatically correct and completely semantically consistent with the source sentence.
> - "4" means the paraphrase has a slight grammatical error, but still maintains the correct semantics, and can also be considered a valuable paraphrase.
> - "3" means the paraphrase has a slight grammatical error and a minor semantic deviation.
> - "2" means the paraphrase has a serious grammatical error and a major semantic deviation, but is still a complete sentence.
> - "1" means the paraphrase is not a complete sentence or is totally irrelevant to the source sentence.
>
> (b) The diversity evaluates whether the paraphrase has diverse expressions compared to the input sentence based on whether the words used are the same or whether the syntactic structure is the same. And the larger the score, the higher the diversity. The detailed guidelines are as follows.
>
> - "5" means the paraphrase has a very different syntax from the source sentence and adopts many new words and phrases.
> - "4" means the paraphrase has a new syntax against the source sentence and revises some original words.
> - "3" means the paraphrase has a new syntax against the source sentence but still adopts original words or phrases.
> - "2" means the paraphrase has the same expression as the source sentence but adopts a few new words or phrases.
> - "1" means the paraphrase is almost identical to the source sentence.

---

### Meta-Review · Area_Chair_dcSz · 2023-09-18

**Recommendation:** 5

**Metareview:**

Reasons to accept
(1) The paper addresses the important issue of syntactically controlled paraphrase generation, specifically focusing on selecting appropriate templates.
(2) The proposed method is intuitive and can be applied to any Syntactically Controlled Paraphrase Generation (SPG) model to enhance its performance.
(3) The paper highlights and addresses the potential problem of low diversity in generated paraphrases when using a quality-based retriever approach to select templates.
(4) Thorough evaluation includes a wide range of baselines, automated metrics, and human evaluations, providing comprehensive insights into the proposed approach's effectiveness, including its role in data augmentation for downstream tasks.

Reasons to Reject:
(1) The paper's proposed method appears to be less effective when used with the SISCP SPG model compared to the performance gains observed for AESOP.
(2) Some parts of the paper are not clearly written, making it challenging for readers to fully understand certain aspects, such as the choice of templates and their role.
(3) The task definition could be clearer, and the choice of exemplar-as-template as a baseline may not align with the typical setup for controlled paraphrase generation tasks.
(4) The lack of certain strong baselines from previous related work and the use of two RoBERTa-based encoders make it challenging to determine whether performance gains are due to the proposed method itself or other factors.
(5) The manual evaluation scores do not include a separate evaluation of syntactic controllability, which is a central aspect of the task.

Two of three reviewers agree that the paper is strong and exciting, however one reviewer gave ambivalent marks for the paper's strength and excitement. The authors also carefully responded to each reviewer's comments and questions by providing additional information, detailed explanations, or new experiments. I have read through the responses from the authors and consider their responses to be reasonable and valid.

---

### Decision · Program_Chairs · 2023-10-07

**Decision:**

Accept-Main

**Comment:**

Reasons to accept
(1) The paper addresses the important issue of syntactically controlled paraphrase generation, specifically focusing on selecting appropriate templates.
(2) The proposed method is intuitive and can be applied to any Syntactically Controlled Paraphrase Generation (SPG) model to enhance its performance.
(3) The paper highlights and addresses the potential problem of low diversity in generated paraphrases when using a quality-based retriever approach to select templates.
(4) Thorough evaluation includes a wide range of baselines, automated metrics, and human evaluations, providing comprehensive insights into the proposed approach's effectiveness, including its role in data augmentation for downstream tasks.

Reasons to Reject:
(1) The paper's proposed method appears to be less effective when used with the SISCP SPG model compared to the performance gains observed for AESOP.
(2) Some parts of the paper are not clearly written, making it challenging for readers to fully understand certain aspects, such as the choice of templates and their role.
(3) The task definition could be clearer, and the choice of exemplar-as-template as a baseline may not align with the typical setup for controlled paraphrase generation tasks.
(4) The lack of certain strong baselines from previous related work and the use of two RoBERTa-based encoders make it challenging to determine whether performance gains are due to the proposed method itself or other factors.
(5) The manual evaluation scores do not include a separate evaluation of syntactic controllability, which is a central aspect of the task.

Two of three reviewers agree that the paper is strong and exciting, however one reviewer gave ambivalent marks for the paper's strength and excitement. The authors also carefully responded to each reviewer's comments and questions by providing additional information, detailed explanations, or new experiments. I have read through the responses from the authors and consider their responses to be reasonable and valid.